# Looking for ELMo's friends: Sentence-Level Pretraining Beyond Language Modeling

## Abstract

Work on the problem of contextualized word representation—the development of reusable neural network components for sentence understanding—has recently seen a surge of progress centered on the unsupervised pretraining task of *language modeling* with methods like ELMo (Peters et al., 2018). This paper contributes the first large-scale systematic study comparing different pretraining tasks in this context, both as complements to language modeling and as potential alternatives. The primary results of the study support the use of language modeling as a pretraining task and set a new state of the art among comparable models using multitask learning with language models. However, a closer look at these results reveals worryingly strong baselines and strikingly varied results across target tasks, suggesting that the widely-used paradigm of pretraining and freezing sentence encoders may not be an ideal platform for further work.

## 1 Introduction

State-of-the-art models for natural language processing (NLP) tasks like translation, question answering, and parsing include components intended to extract representations for the meaning and contents of each input sentence. These *sentence encoder* components are typically trained directly for the target task at hand. This approach can be effective on data rich tasks and yields human performance on some narrowly-defined benchmarks (Rajpurkar et al., 2018; Hassan et al., 2018), but it is tenable only for the few NLP tasks with millions of examples of training data. This has prompted interest in *pretraining* for sentence encoding: There is good reason to believe it should be possible to exploit outside data and training signals to effectively pretrain these encoders, both because they are intended to primarily capture sentence meaning rather than any task-specific skill, and because we have seen dramatic successes with pretraining in the related domains of word embeddings (Mikolov et al., 2013) and image encoders (Zamir et al., 2018).

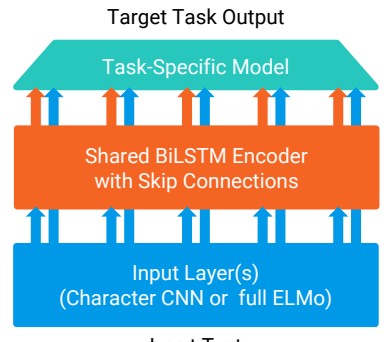

Figure 1: Our common model design: During pretraining, we train the shared encoder and the task-specific model for each pretraining task. We then freeze the shared encoder and train the task-specific model anew for each target evaluation task. Tasks may involve more than one sentence.

More concretely, four recent papers show that pretrained sentence encoders can yield very strong performance on NLP tasks. First, McCann et al. (2017) show that a BiLSTM encoder from a neural machine translation (MT) system can be effectively reused elsewhere. Howard & Ruder (2018), Peters et al. (2018), and Radford et al. (2018) show that various kinds of encoder pretrained in an unsupervised fashion through generative *language modeling* (LM) are effective as well. Each paper uses its own evaluation methods, though, making it unclear which pretraining task is most effective or whether multiple pretraining tasks can be productively combined; in the related setting of sentence-to-vector encoding, multitask learning with multiple *labeled* datasets has yielded a robust state of the art (Subramanian et al., 2018).

This paper attempts to systematically address these questions. We train reusable sentence encoders on 17 different pretraining tasks, several simple baselines, and several combinations of these tasks, all using a single model architecture and procedure for pretraining and transfer, inspired by ELMo. We then evaluate each of these encoders on the nine target language understanding tasks in the GLUE benchmark (Wang et al., 2018), yielding a total of 40 sentence encoders and 360 total trained models. We then measure correlation in performance across target tasks and plot learning curves evaluating the effect of training data volume on each pretraining and target tasks.

Looking to the results of this experiment, we find that language modeling is the most effective single pretraining task we study, and that multitask learning during pretraining can offer further gains and a new state-of-the-art among fixed sentence encoders. We also, however, find reasons to worry that ELMo-style pretraining, in which we pretrain a model and use it on target tasks with no further fine-tuning, is brittle and seriously limiting: (i) Trivial baseline representations do nearly as well as the best pretrained encoders, and the margins between substantially different pretraining tasks can be extremely small. (ii) Different target tasks differ dramatically on what kinds of pretraining they benefit most from, and multitask pretraining is not sufficient to circumvent this problem and offer general-purpose pretrained encoders.

## 2 RELATED WORK

Work toward learning reusable sentence encoders can be traced back at least as far as the multitask model of Collobert et al. (2011), but has seen a recent surge in progress with the successes of CoVe (McCann et al., 2017), ULMFit (Howard & Ruder, 2018), ELMo (Peters et al., 2018), and the Transformer LM (Radford et al., 2018). However, each uses a different model and dataset from the others, so while these works serve as existence proofs that effective reusable sentence encoders are possible, they do not address the question of what task or tasks should be used to create them.

The revival of interest in sentence encoder pretraining is recent enough that relatively little has been done to understand the relative merits of these models, though two exceptions stand out. In unpublished work, Zhang & Bowman (2018) offer an analysis of the relative strengths of translation and language modeling using a single architecture and training dataset. They find that encoders trained as language models reliably uncover the most syntactic structure, even when they are trained on a strict subset of the data used for a comparable translation model. Peters et al. offer a deeper investigation of model design issues for ELMo, showing that all of the standard architectures for sentence encoding can be effectively pretrained with broadly similar performance, and that all learn reasonably good representations of the morphological and syntactic properties of sentences.

There has been a great deal of work on sentence-to-vector encoding, a setting in which the pretrained encoder produces a *fixed-size* vector representation for each input sentence (Dai & Le, 2015; Kiros et al., 2015; Hill et al., 2016; Conneau et al., 2017; Yang et al., 2018). These vectors are potentially useful for tasks that require fast similarity-based matching of sentences, but using them to replace sentence encoders trained in the conventional way on a given target text classification task does not reliably yield state-of-the art performance on that task (Subramanian et al., 2018).

Multitask representation learning in NLP in general has been well studied, and again can be traced back at least as far as Collobert et al. (2011). For example, Luong et al. (2016) show promising results from the combination of translation and parsing, Subramanian et al. (2018) show the benefits of multitask learning in sentence-to-vector encoding, and Bingel & Søgaard (2017) and Changpinyo et al. (2018) offer studies of when multitask learning is helpful for lower-level NLP tasks.

## 3 PRETRAINING TASKS

Our main experiment compares encoders pretrained on a large number of tasks and task combinations, where a *task* is a dataset–objective function pair. This section lists these tasks, which we select either to serve as baselines or because they have shown promise in outside prior work, especially prior work on sentence-to-vector encoding. Appendix A includes additional details on how we implemented some of these tasks, and names tasks we evaluated but left out.

**Random Encoder**  Our primary baseline is equivalent to pretraining on a task with zero examples. Here, we randomly initialize a sentence encoder and use it directly with no further training. This baseline works well, yielding scores far above those of a bag-of-words encoder.[1] This surprising result matches results seen recently with ELMo-like models by Zhang & Bowman (2018) and earlier work on Reservoir Computing. This baseline is especially strong because our model contains a skip connection from the *input* of the shared encoder to its output, allowing the task-specific model to directly see our word representations, or, in experiments where we use a pretrained ELMo model as our input layer, ELMo's contextual word representations.

**GLUE Tasks**  We use the nine tasks included with GLUE as pretraining tasks: acceptability classification with CoLA (Warstadt et al., 2018); binary sentiment classification with SST (Socher et al., 2013); semantic similarity with the MSR Paraphrase Corpus (MRPC; Dolan & Brockett, 2005), the Quora Question Pairs[2] (QQP), and STS-Benchmark (STS; Cer et al., 2017); and textual entailment with the Multi-Genre NLI Corpus (MNLI Williams et al., 2018), RTE 1, 2, 3, and 5 (RTE; Dagan et al., 2006, et seq.), and data from SQuAD (QNLI, Rajpurkar et al., 2016) and the Winograd Schema Challenge (WNLI, Levesque et al., 2011) recast as entailment in the style of White et al. (2017). MNLI is the only task with substantial prior work in this area, as it was found to be highly effective as a pretraining strategy by Conneau et al. (2017) and Subramanian et al. (2018). Other tasks are included to represent a broad sample of labeling schemes commonly used in NLP.

**Language Modeling**  We train language models on two datasets: WikiText-103 (WP, Merity et al., 2017) and 1 Billion Word Language Model Benchmark (BWB, Chelba et al., 2013), which are used by ULMFit (Howard & Ruder, 2018) and ELMo (Peters et al., 2018) respectively.

**Translation**  We train MT models on two datasets: WMT14 English-German (Bojar et al., 2014) and WMT17 English-Russian (Bojar et al., 2017).

**SkipThought**  Our SkipThought model (Kiros et al., 2015; Tang et al., 2017) is a sequence-to-sequence model that reads a sentence from WikiText-103 running text and attempts to decode the following sentence from that text.

**DisSent**  We train our DisSent model (Jernite et al., 2017; Nie et al., 2017) to read two separate clauses that appear in WikiText-103 connected by a discourse marker such as *and*, *but*, or *so* and predict the identity of the discourse marker.

**Reddit**  These models reconstruct comment threads from `reddit.com` using a dataset of about 18M comment–response pairs collected from 2008-2011 by Yang et al. (2018). We consider two settings: A classification task in which the model makes a binary prediction about whether a candidate response is the actual response to a given comment, and a sequence-to-sequence task in the model attempts to generate the true response to a comment.

## 4   MODELS AND TRAINING PROCEDURES

We implement our models using the AllenNLP toolkit (Gardner et al., 2017), aiming to build the simplest architecture that could be reasonably expected to perform well on the target tasks under study.[3] The design of the models roughly follows that used in the GLUE baselines and ELMo.

**Shared Encoder**  The core of our model is a two-layer 1024D bidirectional LSTM. We feed the word representations to the biLSTM and take the sequence of hidden states from the top-level LSTM as the contextual representation. The downstream task-specific model sees both the top-layer hidden states of this model and, through a skip connection, the input representations for each word.

---

[1]This baseline yields 67.4 GLUE, compared to 58.9 with BoW from Wang et al. (2018).

[2] `data.quora.com/First-Quora-Dataset-Release-Question-Pairs`

[3]The code is documented for easy reuse. A link will be added upon acceptance.

**Input Handling and Pretrained ELMo**    All of our models use the pretrained character-level convolutional neural network (CNN) word encoder from ELMo (Peters et al., 2018). This encoder acts as a standard input layer which uses no information beyond the word, and allows us to avoid potentially the difficult issues surrounding unknown word handling in transfer learning.

In some experiments, we use the *full* pretrained ELMo model as an input handler, yielding a form of multitask learning in which the lower layers of the overall model (ELMo) are pretrained on language modeling, and the higher layers (our shared encoder) are pretrained on some additional task or tasks. We choose to use this pretrained model because it represents a larger model with more extensive tuning than we have the resources to produce ourselves. We compare pretraining tasks in this setting to understand how well they *complement* large-scale language model pretraining, and we additionally train our own language models to directly compare between language modeling and other pretraining methods. We follow the standard practice of training a set of scalar weights of ELMo's three layers. We use one set of weights to supply input to the shared encoder, and an additional set for each target task to use in the skip connection. We use only ELMo and not the similarly-situated CoVe, as Wang et al. (2018) showed CoVe to be less effective on the GLUE tasks.

**Evaluation and Per-Task Models**    The GLUE benchmark (Wang et al., 2018) is an open-ended shared task competition and evaluation toolkit for reusable sentence encoders, and we use it as our primary vehicle for evaluation. GLUE is a set of nine classification or regression tasks over sentences and sentence pairs spanning a range of dataset sizes, paired with private test data and an online leaderboard. GLUE offers a larger set of tasks than evaluated by ELMo or CoVe while omitting more expensive paragraph-level tasks, allowing us to evaluate a substantially larger number of experiments with available compute resources.

To evaluate the shared encoder, we use the following procedure: We freeze the pretrained encoder and, for each of the nine tasks in the GLUE benchmark, separately train a target-task model on the representations produced by the encoder. We then evaluate each of these models on the validation or test set of the corresponding task using the standard metric(s) for that task, and report the resulting scores and the overall average GLUE scores, which weight each task equally.

For single-sentence target tasks (CoLA, SST) and sentence-pair tasks with smaller training datasets (MRPC, RTE, WNLI) we train a linear projection over the output states of the shared encoder, max-pool over those projected states, and feed the results to a one-hidden-layer classifier MLP. For smaller sentence pair-tasks, we perform these steps on both sentences and use the *heuristic matching* feature vector $[h_1; h_2; h_1 \cdot h_2; h_1 - h_2]$ in the MLP, following Mou et al. (2016).

For the remaining sentence-pair tasks (MNLI, QNLI, QQP, STS), we use an attention mechanism between all pairs of words, followed by a $512D \times 2$ BiLSTM with max-pooling over time, following the basic mechanism used in BiDAF (Seo et al., 2017). This is followed by heuristic matching and a final MLP, as above. Appendices A and B present additional details on the task specific models.

**Pretraining Task Models**    For pretraining on GLUE tasks, we use the architecture described above, except that we do not use an attention mechanism, as early results indicated that this hurt cross-task transfer performance. For consistency with other experiments when pretraining on a GLUE task, we reinitialize the task-specific parameters between pretraining and target-task training.

Several of the outside (non-GLUE) pretraining tasks involve sentence pair classification. For these, we use the same non-attentive architecture as for the larger GLUE tasks. For LM, to prevent information leakage across directions and LSTM layers, we follow the broad strategy used by ELMo: We train separate forward and backward two-layer LSTM language models, and concatenate the outputs during target task training. For sequence-to-sequence pretraining tasks (MT, SkipThought, Reddit), we use an LSTM decoder with a single layer.

**Multitask Learning**    We also investigate three sets of tasks for multitask pretraining: all GLUE tasks, all outside (non-GLUE) pretraining tasks, and all pretraining tasks. Because ELMo representations are computed with the full context and so cannot be used as the input to downstream unidirectional language models, we exclude language modeling from multitask runs that use ELMo. At each update during multitask learning, we randomly sample a single task with probability proportional to its training data size raised to the power of 0.75. This sampling rate is meant to balance the risks of overfitting small-data tasks and underfitting large ones, and performed best in early exper-

iments. More extensive experiments with methods like this are shown in Appendix C. We perform early stopping based on an unweighted average of the pretraining tasks' validation metrics. For validation metrics like perplexity that decrease from high starting values during training, we include the transformed metric $1 - \frac{m}{250}$ in our average, tuning the constant 250 in early experiments.

**Optimization** We train our models with the AMSGrad optimizer (Reddi et al., 2018)—a variant of Adam (Kingma & Ba, 2015). We perform early stopping at pretraining time and target task training time using the respective dev set performances. Typical experiments, including pretraining one encoder and training the nine associated target-task models, take 1–5 days to complete on an NVIDIA P100 GPU. See Appendix B for more details.

**Hyperparameter Tuning** Appendix B describes our chosen hyperparameter values. As our primary experiment required more than 100 GPU-days on NVIDIA P100 GPUs to run—not counting debugging or learning curves—we did not have the resources for extensive hyperparameter tuning. Instead of carefully tuning our shared and task-specific models on a single pretraining task in a way that might bias results toward that task, we simply chose commonly-used values for most hyperparameters. The choice not to tune limits our ability to diagnose the causes of poor performance when it occurs, and we invite readers to further refine our models using the public code.

## 5 RESULTS

Table 1 shows results on the GLUE dev set for all our pretrained encoders, each with and without the pretrained ELMo BiLSTM layers ($^E$). The *N/A* baselines are untrained encoders with random intialization. The *Single-Task* baselines are aggregations of results from nine GLUE runs: The result in this row for a given GLUE task uses the encoder pretrained on only that task. For consistency with other runs, we treat the pretraining task and the target task as two separate tasks in all cases (including here) and give them separate task-specific parameters, despite the fact that they use identical data. We use $^S$ and $^C$ to distinguish the sequence-to-sequence and classification versions of the Reddit task, respectively.

To comply with GLUE's limits on test set access, we evaluated only three of our pretrained encoders on test data. These reflect our best models with and without the use of the pretrained ELMo encoder, and with and without the use of GLUE data during pretraining. For discussion of our limited hyperparameter tuning, see above. For roughly-comparable GLUE results in prior work, see Wang et al. (2018) or `https://www.gluebenchmark.com`; we omit them here in the interest of space. The limited size of a US Letter page prevent us from including these baselines in this table. As of writing, the best test result using a comparable frozen pretrained encoder is 68.9 from Wang et al. (2018) for a model similar to our GLUE$^E$ multitask model, and the best overall result is 72.8 from Radford et al. (2018) with a model that is fine-tuned in its entirety for each target task.

While not feasible to run each setting multiple times, we estimate the variance of the GLUE score by re-running the random encoder and MNLI pretraining setups with and without ELMo with different random seeds. Across five runs, we recorded $\sigma = 0.4$ for the random encoder (*N/A* in table), and $\sigma = 0.2$ for MNLI$^E$. This variation is substantial but not so high as to render results meaningless.

For the explicitly adversarial WNLI dataset (based on the Winograd Schema Challenge; Levesque et al., 2011), only one of our models reached even the *most frequent class* performance of 56.3. In computing average and test set performances, we replace model predictions with the most frequent label to simulate the better performance achievable by choosing not to model that task.

Looking to other target tasks, the grammar-related CoLA task benefits dramatically from ELMo pretraining: The best result without language model pretraining is *less than half* the result achieved with such pretraining. In contrast, the meaning-oriented textual similarity benchmark STS sees good results with several kinds of pretraining, but does not benefit substantially from the use of ELMo.

Comparing pretraining tasks in isolation without ELMo, language modeling performs best, followed by MNLI. The remaining pretraining tasks yield performance near that of the random baseline. Even when training directly on each target task (*Single-Task* in table), we get less than a one point gain over this simple baseline. Adding ELMo yielded improvements in performance across all pretraining

Table 1: GLUE results, using the development sets except where noted. $^E$: ELMo used as input layer. $^C$ and $^S$ distinguish the two variants of the Reddit task. **Bold** results are the best overall; underlined results are the best without ELMo. See text for discussion of WNLI results (*).

| Pretr. | Avg | CoLA | SST | MRPC | QQP | STS | MNLI | QNLI | RTE | WNLI |
|---|---|---|---|---|---|---|---|---|---|---|
| | | | | Baselines | | | | | | |
| **N/A** | 68.2 | 16.9 | 84.3 | 77.7/85.6 | 83.0/80.6 | 81.7/82.6 | 73.9 | 79.6 | 57.0 | 31.0* |
| **Single-Task** | 69.1 | 21.3 | 89.0 | 77.2/84.7 | 84.7/81.9 | 81.4/82.2 | 74.8 | 78.8 | 56.0 | 11.3* |
| **N/A**$^E$ | 70.5 | 38.5 | 87.7 | 79.9/86.5 | 86.7/83.4 | 80.8/82.1 | 75.6 | 79.6 | 61.7 | 33.8* |
| **Single-Task**$^E$ | 71.2 | 39.4 | **90.6** | 77.5/84.4 | 86.4/82.4 | 79.9/80.6 | 75.6 | 78.0 | 55.6 | 11.3* |
| | | | | GLUE Tasks as Pretraining Tasks | | | | | | |
| **CoLA** | 68.2 | 21.3 | 85.7 | 75.0/83.7 | 85.7/82.4 | 79.0/80.3 | 72.7 | 78.4 | 56.3 | 15.5* |
| **MNLI** | 69.1 | 16.7 | 88.2 | 78.9/85.2 | 84.5/81.5 | 81.8/82.6 | 74.8 | 79.6 | 58.8 | 36.6* |
| **MRPC** | 68.2 | 16.4 | 85.6 | 77.2/84.7 | 84.4/81.8 | 81.2/82.2 | 73.6 | 79.3 | 56.7 | 22.5* |
| **QNLI** | 67.9 | 15.6 | 84.2 | 76.5/84.2 | 84.3/81.4 | 80.6/81.8 | 73.4 | 78.8 | 58.8 | **56.3** |
| **QQP** | 68.0 | 14.7 | 86.1 | 77.2/84.5 | 84.7/81.9 | 81.1/82.0 | 73.7 | 78.2 | 57.0 | 45.1* |
| **RTE** | 68.1 | 18.1 | 83.9 | 77.5/85.4 | 83.9/81.2 | 81.2/82.2 | 74.1 | 79.1 | 56.0 | 39.4* |
| **SST** | 68.6 | 16.4 | 89.0 | 76.0/84.2 | 84.4/81.6 | 80.6/81.4 | 73.9 | 78.5 | 58.8 | 19.7* |
| **STS** | 67.7 | 14.1 | 84.6 | 77.9/85.3 | 81.7/79.2 | 81.4/82.2 | 73.6 | 79.3 | 57.4 | 43.7* |
| **WNLI** | 68.0 | 16.3 | 84.3 | 76.5/84.6 | 83.0/80.5 | 81.6/82.5 | 73.6 | 78.8 | 58.1 | 11.3* |
| **CoLA**$^E$ | 71.1 | 39.4 | 87.3 | 77.5/85.2 | 86.5/83.0 | 78.8/80.2 | 74.2 | 78.2 | 59.2 | 33.8* |
| **MNLI**$^E$ | 72.1 | 38.9 | 89.0 | 80.9/86.9 | 86.1/82.7 | 81.3/82.5 | 75.6 | 79.7 | 58.8 | 16.9* |
| **MRPC**$^E$ | 71.3 | 40.0 | 88.4 | 77.5/84.4 | 86.4/82.7 | 79.5/80.6 | 74.9 | 78.4 | 58.1 | 54.9* |
| **QNLI**$^E$ | 71.2 | 37.2 | 88.3 | 81.1/86.9 | 85.5/81.7 | 78.9/80.1 | 74.7 | 78.0 | 58.8 | 22.5* |
| **QQP**$^E$ | 70.8 | 34.3 | 88.6 | 79.4/85.7 | 86.4/82.4 | 81.1/82.1 | 74.3 | 78.1 | 56.7 | 38.0* |
| **RTE**$^E$ | 71.2 | 38.5 | 87.7 | 81.1/87.3 | 86.6/83.2 | 80.1/81.1 | 74.6 | 78.0 | 55.6 | 32.4* |
| **SST**$^E$ | 71.2 | 38.8 | **90.6** | 80.4/86.8 | 87.0/83.5 | 79.4/81.0 | 74.3 | 77.8 | 53.8 | 43.7* |
| **STS**$^E$ | 71.6 | 39.9 | 88.4 | 79.9/86.4 | 86.7/83.3 | 79.9/80.6 | 74.3 | 78.6 | 58.5 | 26.8* |
| **WNLI**$^E$ | 70.9 | 38.4 | 88.6 | 78.4/85.9 | 86.3/82.8 | 79.1/80.0 | 73.9 | 77.9 | 57.0 | 11.3* |
| | | | | Outside Pretraining Tasks | | | | | | |
| **DisSent WP** | 68.6 | 18.3 | 86.6 | 79.9/86.0 | 85.3/82.0 | 79.5/80.5 | 73.4 | 79.1 | 56.7 | 42.3* |
| **LM WP** | 70.1 | 30.8 | 85.7 | 76.2/84.2 | 86.2/82.9 | 79.2/80.2 | 74.0 | 79.4 | 60.3 | 25.4* |
| **LM BWB** | 70.4 | 30.7 | 86.8 | 79.9/86.2 | 86.3/83.2 | 80.7/81.4 | 74.2 | 79.0 | 57.4 | 47.9* |
| **MT En-De** | 68.1 | 16.7 | 85.4 | 77.9/84.9 | 83.8/80.5 | 82.4/82.9 | 73.5 | 79.6 | 55.6 | 22.5* |
| **MT En-Ru** | 68.4 | 16.8 | 85.1 | 79.4/86.2 | 84.1/81.2 | 82.7/83.2 | 74.1 | 79.1 | 56.0 | 26.8* |
| **Reddit**$^C$ | 68.2 | 15.9 | 85.0 | 78.9/84.5 | 85.0/82.0 | 82.3/82.8 | 73.5 | 78.9 | 56.0 | 35.2* |
| **Reddit**$^S$ | 66.9 | 15.3 | 82.3 | 76.5/84.6 | 81.9/79.2 | 81.5/81.9 | 72.7 | 76.8 | 55.6 | 53.5* |
| **SkipThought** | 68.7 | 16.0 | 84.9 | 77.5/85.0 | 83.5/80.7 | 81.1/81.5 | 73.3 | 79.1 | 63.9 | 49.3* |
| **DisSent WP**$^E$ | 71.9 | 39.9 | 87.6 | **81.9**/87.2 | 85.8/82.3 | 79.0/80.7 | 74.6 | 79.1 | 61.4 | 23.9* |
| **MT En-De**$^E$ | 72.1 | 40.1 | 87.8 | 79.9/86.6 | 86.4/83.2 | 81.8/82.4 | 75.9 | 79.4 | 58.8 | 31.0* |
| **MT En-Ru**$^E$ | 70.4 | **41.0** | 86.8 | 76.5/85.0 | 82.5/76.3 | 81.4/81.5 | 70.1 | 77.3 | 60.3 | 45.1* |
| **Reddit**$^{C\,E}$ | 71.8 | 37.0 | 88.1 | 79.9/86.8 | 86.1/83.0 | 81.7/82.6 | 75.3 | 78.7 | 61.0 | 25.4* |
| **Reddit**$^{S\,E}$ | 71.0 | 38.5 | 87.7 | 77.2/85.0 | 85.4/82.1 | 80.9/81.7 | 74.2 | 79.3 | 56.7 | 21.1* |
| **SkipThought**$^E$ | 71.7 | 40.6 | 87.7 | 79.7/86.5 | 85.2/82.1 | 81.0/81.7 | 75.0 | 79.1 | 58.1 | 52.1* |
| | | | | Multitask Pretraining | | | | | | |
| **GLUE** | 68.9 | 15.4 | 89.9 | 78.9/86.3 | 82.6/79.9 | 82.9/83.5 | 74.9 | 78.9 | 57.8 | 38.0* |
| **Outside** | 69.9 | 30.6 | 87.0 | 81.1/**87.6** | 86.0/82.2 | 79.9/80.6 | 72.8 | 78.9 | 54.9 | 22.5* |
| **All** | 70.4 | 33.2 | 88.2 | 78.9/85.9 | 85.5/81.8 | 79.7/80.0 | 73.9 | 78.7 | 57.4 | 33.8* |
| **GLUE**$^E$ | 72.1 | 33.8 | 90.5 | 81.1/87.4 | 86.6/83.0 | 82.1/83.3 | **76.2** | 79.2 | 61.4 | 42.3* |
| **Outside**$^E$ | **72.4** | 39.4 | 88.8 | 80.6/86.8 | **87.1/84.1** | **83.2/83.9** | 75.9 | **80.9** | 57.8 | 22.5* |
| **All**$^E$ | 72.2 | 37.9 | 89.6 | 79.2/86.4 | 86.0/82.8 | 81.6/82.5 | 76.1 | 80.2 | 60.3 | 31.0* |
| | | | | *Test Set Results* | | | | | | |
| **LM BWB** | 66.5 | 29.1 | 86.9 | 75.0/82.1 | 82.7/63.3 | 74.0/73.1 | 73.4 | 68.0 | 51.3 | 65.1 |
| **All** | 68.5 | 36.3 | 88.9 | 77.7/84.8 | 82.7/63.6 | 77.8/76.7 | 75.3 | 66.2 | 53.2 | 65.1 |
| **Outside**$^E$ | 69.7 | 34.5 | 89.5 | 78.2/84.8 | 83.6/64.3 | 77.5/76.0 | 75.4 | 74.8 | 55.6 | 65.1 |

Table 2: Pearson correlations between performances on different target tasks, measured over all runs reported in Table 1. The *Avg* column shows the correlation between an individual task's performance for some pretraining and the overall GLUE score for that run. For tasks with multiple metrics, we note the metric used here in the row title. Negative correlations are underlined.

| Task | Avg | CoLA | SST | MRPC | STS | QQP | MNLI | QNLI | RTE | WNLI |
|------|------|------|------|------|------|------|------|------|------|------|
| **CoLA** | 0.95 | 1.00 | | | | | | | | |
| **SST** | 0.75 | 0.62 | 1.00 | | | | | | | |
| **MRPC** | 0.71 | 0.61 | 0.58 | 1.00 | | | | | | |
| **STS** | -0.21 | -0.38 | -0.06 | 0.06 | 1.00 | | | | | |
| **QQP** | 0.74 | 0.75 | 0.56 | 0.37 | -0.37 | 1.00 | | | | |
| **MNLI** | 0.80 | 0.65 | 0.72 | 0.70 | 0.26 | 0.51 | 1.00 | | | |
| **QNLI** | 0.18 | 0.03 | 0.12 | 0.15 | 0.34 | 0.12 | 0.44 | 1.00 | | |
| **RTE** | 0.35 | 0.21 | 0.18 | 0.20 | 0.04 | 0.15 | 0.43 | 0.45 | 1.00 | |
| **WNLI** | -0.25 | -0.23 | -0.26 | -0.11 | 0.18 | -0.28 | -0.15 | -0.25 | -0.09 | 1.00 |

tasks. MNLI and English–German translation perform best in this setting, with SkipThought, Reddit classification, and DisSent also outperforming the ELMo-augmented random baseline.

With ELMo, a multitask model performs best, but without it, all three multitask models are tied or outperformed by models trained on one of their constituent tasks, suggesting that our approach to multitask learning is not reliably able to produce models that productively use the knowledge taught by each training task. However, of the two non-ELMo models that perform best on the development data, the multitask model generalizes better than the single-task model on test data for tasks like STS where the test set contains new out-of-domain data.

## 6 ANALYSIS AND DISCUSSION

**Cross-Task Correlations**   Table 2 presents an alternative view of the results of the main experiment (Table 1): The table shows the correlations between pairs of tasks over the space of pretrained encoders. These reflect the degree to which knowing the performance of one target task with some encoder will allow us to predict the performance of the other target task with that same encoder.

Many correlations are low, suggesting that different tasks benefit from different forms of pretraining to a substantial degree, and mirroring the observation that no one pretraining task yields good performance on all target tasks. As noted above, the models that tended to perform best overall also overfit the WNLI training set most, leading to a negative correlation between WNLI and overall GLUE score. STS also shows a negative correlation, likely due to the observation that it does not benefit from ELMo pretraining. In contrast, CoLA shows a strong 0.93 correlation with the overall GLUE scores, but has weak or negative correlations with many tasks—the use of ELMo or LM pretraining dramatically improves CoLA performance, but most other forms of pretraining have little effect.

**Learning Curves**   Figure 2 shows two types of learning curves. The first set measures performance on the overall GLUE metric for encoders trained to convergence on each pretraining task with varying amounts of data. The second set focuses on three pretrained encoders and measures performance on each GLUE target task separately with varying amounts of target task data.

Looking at pretraining tasks in isolation (top left), most tasks improve slightly as the amount of pretraining data increases, with the LM and MT tasks showing the most promising combination of slope and maximum performance. Combining these pretraining tasks with ELMo (top right) yields a less interpretable result: the relationship between training data volume and performance becomes weaker, and some of the best results reported in this paper are achieved by models that combine pretrained ELMo with restricted-data versions of other pretraining tasks like MNLI and QQP.

Looking at target task performance as target task training data volume varies, we see that all tasks benefit from increasing data quantities, with no obvious diminishing returns, and that most tasks see a constant improvement in performance across data volumes from the use of pretraining, either with ELMo (center) or with multitask learning (right).

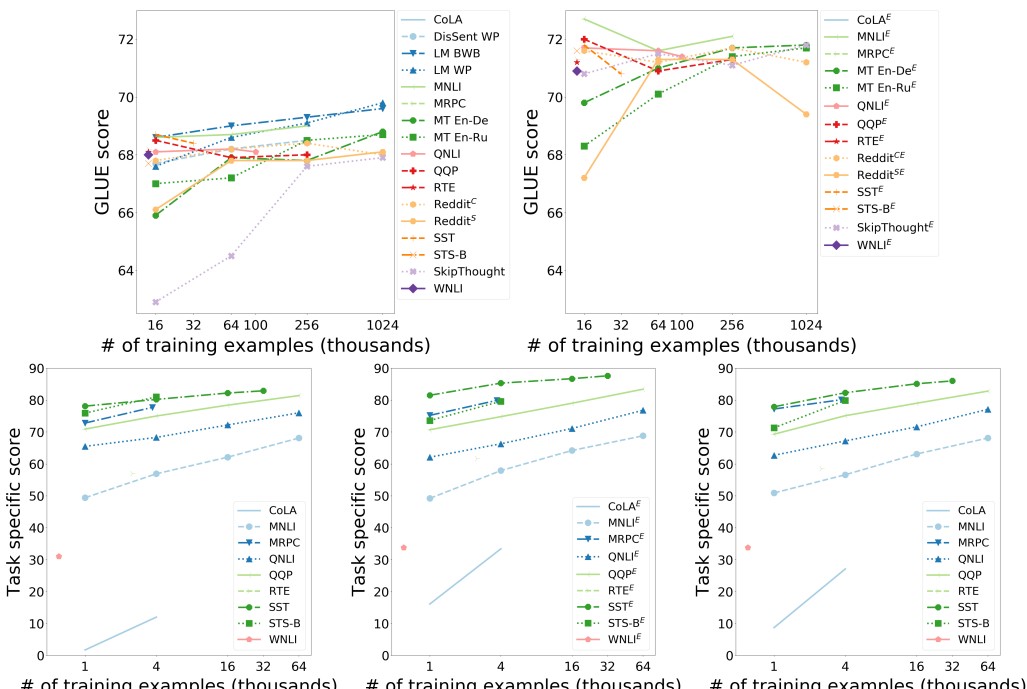

Figure 2: Top: Pretraining learning curves for GLUE score without ELMo (left) and with ELMo (right). Bottom: Target-task training Learning curves for each GLUE task with three encoders: The random encoder without ELMo (left) and with it (center), and Outside MTL without ELMo (right).

**Results on the GLUE Diagnostic Set**   From GLUE's auxiliary diagnostic analysis dataset, we find that ELMo and other forms of unsupervised pretraining helps on examples that involve world knowledge and lexical-semantic knowledge, and less so on examples that highlight complex sentence structures. See Table 6 in Appendix D for more details.

## 7   CONCLUSIONS

This paper presents a systematic comparison of tasks and task-combinations for the pretraining of sentence-level BiLSTM encoders like those seen in ELMo and CoVe. With 40 pretraining tasks and task combinations (not counting many more ruled out early) and nine target tasks, this represents a far more comprehensive study than any seen on this problem to date.

Our chief positive results are perhaps unsurprising: Language modeling works well as a pretraining task, and no other single task is consistently better. Multitask pretraining can produce results better than any single task can, and sets a new state-of-the-art among comparable models.  Target task performance continues to improve with the addition of more language model data, even at large scales, suggesting that further work scaling up language model pretraining is warranted.

However, a closer look at our results suggests that the pretrain-and-freeze paradigm that underlies ELMo and CoVe might not be a sound platform for future work: Some trivial baselines do strikingly well, the margins between pretraining tasks are small, and some pretraining configurations (such as $\text{MNLI}^E$) yield better performance with *less* data.  This suggests that we may be nearing an upper bound on the performance that can be reached with methods like these.

In addition, different tasks benefit from different forms of pretraining to a striking degree—with correlations between target tasks often low or negative—and multitask pretraining tasks fail to reliably produce models better than their best individual components.  This suggests that if truly general-purpose sentence encoders are possible, our current methods cannot produce them.

While further work on language modeling seems straightforward and worthwhile, the author(s) of this paper believe that the future of this line of work will require a better understanding of the ways in which neural network target task models *can* benefit from outside knowledge and data, and new methods for pretraining and transfer learning to allow them to do so.

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

## A ADDITIONAL PRETRAINING TASK DETAILS

**DisSent**   To extract discourse model examples from the WikiText-103 corpus (Merity et al., 2017), we follow the procedure described in Nie et al. (2017) by extracting clause-pairs that follow specific dependency relationships within the corpus (see Figure 4 in Nie et al., 2017). We use the Stanford Parser (Chen & Manning, 2014) distributed in Stanford CoreNLP version 3.9.1 to identify the relevant dependency arcs.

**Reddit Response Prediction**   The Reddit classification task requires a model to select which of two candidate replies to a comment is correct. Since the dataset from Yang et al. (2018) contains only real comment-reply pairs, we select an incorrect distractor reply for each correct reply by permuting each minibatch.

**Alternative Tasks**   Any large-scale comparison like the one attempted in this paper is inevitably incomplete. Among the thousands of publicly available NLP datasets, we also performed initial trial experiments on several datasets for which we were not able to reach development-set performance above that of the random encoder baseline in any setting. These include image-caption matching with MSCOCO (Lin et al., 2014), following Kiela et al. (2018); the small-to-medium-data text-understanding tasks collected in NLI format by Poliak et al. (2018); ordinal common sense inference (Zhang et al., 2017); POS tagging on the Penn Treebank (Marcus et al., 1993); and supertagging on CCGBank (Hockenmaier & Steedman, 2007).

## B HYPERPARAMETERS AND OPTIMIZATION DETAILS

See Section 4 for general comments on hyperparameter tuning.

**Validation**   We evaluate on the validation set for the current training task or tasks every 1,000 steps, except where noted otherwise for small-data target tasks. During multitask learning, we multiply this interval by the number of tasks, evaluating every 9,000 steps during GLUE multitask training, for example.

**Optimizer**  We use AMSGrad (Reddi et al., 2018). During pretraining, we use a learning rate of 1e-4 for classification and regression tasks, and 1e-3 for text generation tasks. During target-task training, we use a learning rate of 3e-4 for all tasks.

**Learning Rate Decay**  We multiply the learning rate by 0.5 whenever validation performance fails to improve for more than 4 validation checks. We stop training if the learning rate falls below 1e-6.

**Early Stopping**  We maintain a saved checkpoint reflecting the best validation result seen so far. We stop training if we see no improvement after more than 20 validation checks. After training, we use the last saved checkpoint.

**Regularization**  We apply dropout with a drop rate of 0.2 after the input layer (the character CNN or ELMo), after each LSTM layer, and after each MLP layer in the task-specific classifier or regressor. For small-data target tasks, we increase MLP dropout to 0.4 during target-task training.

**Preprocessing**  We use Moses tokenizer for encoder inputs, and set a maximum sequence length of 40 tokens. There is no input vocabulary, as we use ELMo's character-based input layer.

For English text generation tasks, we use the Moses tokenizer to tokenize our data, but use a word-level output vocabulary of 20,000 types for tasks that require text generation. For translation tasks, we use BPE tokenization with a vocabulary of 20,000 types. For all sequence-to-sequence tasks we train word embeddings on the decoder side.

**Target-Task-Specific Parameters**  To ensure that baseline performance for each target task is competitive, we find it necessary to use slightly different models and training regimes for larger and smaller target tasks. We used partially-heuristic tuning to separate GLUE tasks into big-, medium- and small-data groups, giving each group its own heuristically chosen task-specific model specifications. Exact values are shown in Table 3.

Table 3: Hyperparameter settings for target-task models and target-task training. Attention is always disabled when *pretraining* on GLUE tasks. STS has a relatively small training set, but consistently patterns with the larger tasks in its behavior.

|  | Small-Data RTE, WNLI | Medium-Data CoLA, SST, MRPC | Large-Data STS, QQP, MNLI, QNLI |
|---|---|---|---|
| Steps btw. validations | 100 | 100 | 1000 |
| Attention | N | N | Y |
| Classifier dropout rate | 0.4 | 0.2 | 0.2 |
| Classifier hidden dim. | 128 | 256 | 512 |
| Max pool projection dim. | 128 | 256 | 512 |

**Sequence-to-Sequence Models**  We found attention to be helpful for the SkipThought and Reddit pretraining tasks but not for machine translation, and report results for these configurations. We use the max-pooled output of the encoder to initialize the hidden state of the decoder, and the size of this hidden state is equal to the size of the output of our shared encoder. We reduce the dimension of the output of the decoder by half via a linear projection before the output softmax layer.

## C  MULTITASK LEARNING METHODS

Our multitask learning experiments have three somewhat distinctive properties: (i) We mix tasks with very different amounts of training data—at the extreme, under 1,000 examples for WNLI, and over 1,000,000,000 examples from LM BWB. (ii) Our goal is to optimize the quality of the shared encoder, not the performance of any one of the tasks in the multitask mix. (iii) We mix a relatively large number of tasks, up to eighteen at once in some conditions. These conditions make it challenging but important to avoid overfitting or underfitting any of our tasks.

Relatively little work has been done on this problem, so we conduct a small experiment here. All our experiments use the basic paradigm of randomly sampling a new task to train on at each step, and we experiment with two hyperparameters that can be used to control over- and underfitting: The probability with which we sample each task and the weight with which we scale the loss for each task. Our experiments follow the setup in Appendix B, and do not use the ELMo BiLSTM.

**Task Sampling** We consider several approaches to determine the probability with which to sample a task during training, generally making this probability a function of the amount of data available for the task. For task $i$ with training set size $N_i$, the probability is $p_i = f(N_i)/\sum_j f(N_j)$, where $f(N_i) = 1$ (Uniform), $N_i$ (Proportional), $log(N_i)$ (Log Proportional), or $N_i^a$ (Power $a$) where $a$ is a constant.

**Loss Scaling** At each update, we scale the loss of a task with weight $w_i = f(N_i)/max_j f(N_j)$, where $f(N_i) = 1$ (Uniform), $N_j$ (Proportional), or $N_j^a$ (Power $a$).

**Experiments** For task sampling, we run experiments with multitask learning on the full set of nine GLUE tasks, as well as three subsets: single sentence tasks (S1: SST, CoLA), similarity and paraphrase tasks (S2: MRPC, STS, QQP), and inference tasks (S3: WNLI, QNLI, MNLI, RTE). The results are shown in Table 4.

We also experiment with several *combinations* of task sampling and loss scaling methods, using only the full set of GLUE tasks. The results are shown in Table 5.

While no combination of methods consistently offers dramatically better performance than any other, we observe that it is generally better to apply only one of non-uniform sampling and non-uniform loss scaling at a time rather than apply both simultaneously, as they provide roughly the same effect. Following encouraging results from earlier pilot experiments, we use power 0.75 task sampling and uniform loss scaling in the multitask learning experiments shown in Table 1.

Table 4: Comparison of sampling methods on four subsets of GLUE using uniform loss scaling. The reported scores are averages of the development set results achieved for each task after early stopping. Results in **bold** are the best within each set.

|  | Pretraining Tasks | | | |
| --- | --- | --- | --- | --- |
| Sampling | GLUE | S1 | S2 | S3 |
| **Uniform** | 69.1 | 53.7 | 82.1 | 31.7 |
| **Proportional** | **69.8** | 52.0 | 83.1 | 36.6 |
| **Log Proportional** | 68.8 | 54.3 | 82.9 | 31.2 |
| **Power 0.75** | 69.3 | 51.1 | 82.7 | **37.9** |
| **Power 0.66** | 69.0 | 53.4 | 82.8 | 35.5 |
| **Power 0.5** | 69.1 | **55.6** | **83.3** | 35.9 |

Table 5: Combinations of sampling and loss scaling methods on GLUE tasks. Results in **bold** are tied for best overall GLUE score.

|  | Loss Scaling | | |
| --- | --- | --- | --- |
| Sampling | Uniform | Proportional | Power 0.75 |
| **Uniform** | 69.1 | **69.7** | **69.8** |
| **Proportional** | **69.8** | 69.4 | 69.6 |
| **Log Proportional** | 68.8 | 68.9 | 68.9 |
| **Power 0.75** | 69.3 | 69.1 | 69.0 |

# D  DIAGNOSTIC SET RESULTS

Table 6, below, shows results on the four coarse-grained categories of the GLUE diagnostic set for all our pretraining experiments. This set consists of about 1000 expert-constructed examples in NLI

Table 6: GLUE diagnostic set results, reported as $R_3$ correlation coefficients ($\times 100$), which standardizes the score of random guessing by an uninformed model at roughly 0. Human performance on the overall diagnostic set is roughly 80. Results in **bold** are the best overall, and underlined results are the best without ELMo.

| Pretr. | Knowledge | Lexical Semantics | Logic | Predicate/Argument Str. |
|---|---|---|---|---|
| **Baselines** | | | | |
| **N/A** | 17.6 | 19.6 | 12.5 | 26.9 |
| **N/A**[E] | 19.2 | 22.9 | 9.8 | 25.5 |
| **GLUE Tasks as Pretraining Tasks** | | | | |
| **CoLA** | 15.3 | 24.2 | 14.9 | **31.7** |
| **MNLI** | 16.4 | 20.4 | **17.7** | 29.9 |
| **MRPC** | 16.0 | **25.2** | 12.6 | 26.4 |
| **QNLI** | 13.6 | 21.3 | 12.2 | 28.0 |
| **QQP** | 12.8 | 22.5 | 12.9 | 30.8 |
| **RTE** | 16.3 | 23.1 | 14.5 | 28.8 |
| **SST** | 16.1 | 24.8 | 16.5 | 28.7 |
| **STS** | 16.5 | 20.2 | 13.0 | 27.1 |
| **WNLI** | 18.8 | 19.5 | 13.9 | 29.1 |
| **CoLA**[E] | 17.2 | 21.6 | 9.2 | 27.3 |
| **MNLI**[E] | 17.0 | 23.2 | 14.4 | 23.9 |
| **MRPC**[E] | 11.8 | 20.5 | 12.1 | 27.4 |
| **QNLI**[E] | 17.4 | 24.1 | 10.7 | 30.2 |
| **QQP**[E] | 17.5 | 16.0 | 9.9 | 30.5 |
| **RTE**[E] | 18.0 | 20.2 | 8.7 | 28.0 |
| **SST**[E] | 19.4 | 20.5 | 9.7 | 28.5 |
| **STS**[E] | 18.0 | 18.4 | 9.1 | 25.5 |
| **WNLI**[E] | 16.5 | 19.8 | 7.3 | 25.2 |
| **Outside Pretraining Tasks** | | | | |
| **DisSent WP** | 18.5 | 24.2 | 15.4 | 27.8 |
| **LM WP** | 14.9 | 16.6 | 9.4 | 23.0 |
| **LM BWB** | 15.8 | 19.4 | 9.1 | 23.9 |
| **MT En-De** | 13.4 | 24.6 | 14.8 | 30.1 |
| **MT En-Ru** | 13.4 | 24.6 | 14.8 | 30.1 |
| **Reddit**[C] | 13.6 | 20.3 | 12.7 | 28.6 |
| **Reddit**[S] | 13.9 | 20.4 | 14.1 | 26.0 |
| **SkipThought** | 15.1 | 22.0 | 13.7 | 27.9 |
| **DisSent WP**[E] | 16.3 | 23.0 | 11.6 | 26.5 |
| **MT En-De**[E] | 19.2 | 21.0 | 13.5 | 29.7 |
| **MT En-Ru**[E] | 20.0 | 20.1 | 11.9 | 21.4 |
| **Reddit**[CE] | 19.4 | 20.7 | 11.9 | 29.2 |
| **Reddit**[SE] | 14.7 | 22.3 | 15.0 | 29.0 |
| **SkipThought**[E] | 20.5 | 18.5 | 10.4 | 26.8 |
| **Multitask Pretraining** | | | | |
| **All** | 16.3 | 21.4 | 11.2 | 28.0 |
| **GLUE** | 12.5 | 21.4 | 15.0 | 30.1 |
| **Outside** | 14.5 | 19.7 | 13.1 | 26.2 |
| **All**[E] | 13.8 | 18.4 | 10.8 | 26.7 |
| **GLUE**[E] | **20.6** | 22.1 | 14.7 | 25.3 |
| **Outside**[E] | 15.7 | 23.7 | 12.6 | 29.0 |

format meant to isolate a range of relevant phenomena. Results use the target task classifier trained on the MNLI training set.

No model achieves performance anywhere close to human-level performance, suggesting that *either* none of our pretrained models extract features that are suitable for robust reasoning over text, or that

the MNLI training set and the MNLI target-task model are not able to exploit any such features that exist. See Section 6 for further discussion.

While no model achieves near-human performance, the use of ELMo and other forms of unsupervised pretraining appears to be helpful on examples that highlight world knowledge and lexical-semantic knowledge, and less so on examples that highlight complex logical reasoning patterns or alternations in sentence structure. This relative weakness on sentence structure is somewhat surprising given the finding in Zhang & Bowman (2018) that language model pretraining is helpful for tasks involving sentence structure.

