# OpenReview forum: "Looking for ELMo's friends: Sentence-Level Pretraining Beyond Language Modeling"
_ICLR.cc/2019/Conference_

### Official Review · AnonReviewer1 · 2018-11-02
**Valuable systematic study of pre-training tasks' influence on downstream task performance**

**Rating:** 8
**Confidence:** 4

**Review:**

This paper presents a thorough and systematic study of the effect of pre-training over various NLP tasks on the GLUE multi-task learning evaluation suite, including an examination of the effect of language model-based pre-training using ELMo. The main conclusion is that both single-task and LM-based pre-training helps in most situations, but the gain is often not large, and not consistent across all GLUE tasks.

This paper represents an impressive amount of experimentation. The study and the experimental results will be useful and interesting to the community. The result that some tasks' performance are negatively correlated with each other is surprising. The paper is clearly written.

One clarification question I have is about what the "Single-task" pre-training means. The paper seems to suggest that it consists of pre-training a model on the same task on which it is later evaluated. I'm confused by what this means, and how this is different from just training on that task.

---

> ### Author Response · Authors · 2018-11-08
> **Author response**
>
> Thanks! We agree with your overall assessment. (I guess that’s not surprising...)
>
> The single-task baselines are a bit confusing, and we’ll clarify that point in an update shortly.
>
> As you describe, we pretrain a model on the same single task that we later evaluate it on. The tricky point here is that we follow the exact same training procedure here as in the other rows of that table, to make sure we’re isolating the effect of the data and not the training procedure. In that procedure, we pretrain an encoder, freeze it, then initialize and train a new task-specific model, so we wind up training two different task-specific models: one (always without attention) that’s used only during encoder pretraining, and another (always with attention) that’s trained on the frozen encoder and then evaluated.
>
> This is fairly complex, but it allows us to make the comparisons we want to make. The single-task baselines in the original GLUE paper are trained in a single pass with a single target-task model, so those runs aren’t precisely comparable to ours.

---

### Official Review · AnonReviewer2 · 2018-11-02
**This paper presents an extremely comprehensive comparison of sentence representation methods.**

**Rating:** 7
**Confidence:** 4

**Review:**

Only a handful of NLP tasks have an ample amount of labeled data to get state-of-the-art results without using any form of transfer learning. Training sentence representation in an unsupervised manner is hence crucial for real-world NLP applications.
Contextualized word representations have gained a lot of interest in recent years and the NLP and ML community could benefit from such detailed comparison of such methods.

This paper's biggest strength is the experimental setting.  The authors cover a lot of ground in comparing a lot of the recent work, both qualitatively and quantitatively -- there are a lot of experiments.
I do understand the computational limitations of the authors (as they mention on HYPERPARAMETER TUINING) and I do agree with their statement “ The choice not to tune limits our ability to diagnose the causes of poor performance when it occurs”.
Extensive hyper-parameter tuning can make a substantial different when dealing with NN models, maybe the authors should have considered dropping some of the tasks (the article has more than enough IMHO) and focus on a smaller sub set of tasks with proper hyper-parameter tuning.
Table 2 is very interesting, the results suggesting that we are indeed very far from fully robust sentence representation method.

---

> ### Author Response · Authors · 2018-11-08
> **Author response**
>
> Thanks!
>
> As you note, tuning is a real limitation that we acknowledge in the paper, and it represents an unavoidable trade-off. By training a lightly-regularized parameter-rich model under training conditions that are generally known to work well, we intend to make a rough but informative comparison possible between data-rich pretraining objectives.
>
> Our lack of substantial hyperparameter tuning means that small differences across pretraining settings are unlikely to be meaningful, and likely means that our performance numbers for our lowest-data pretraining tasks are weaker than they would otherwise be, but we still believe that our overall conclusions are both correct and informative as-is.

---

### Official Review · AnonReviewer3 · 2018-11-06
**Many experiments on a fast-moving field, without clear conclusion**

**Rating:** 5
**Confidence:** 3

**Review:**

The work presented in this paper relates to the impact of the dataset on the performance of contextual embedding (namely ELMO in this paper) on many downstream tasks, including GLUE tasks, but also alternative NLP tasks.

The work is focused on experiments, and draws several conclusions that are interesting, mostly around the amount of gain one can expect and the fact that the choice of the dataset is task-dependent.

One of the issue is that the authors if seems to believe that ELMO is the best contextual language model. The field is moving so quickly that the experiments might become invalid pretty soon (e.g. see BERT model referenced below).

Finally, the analysis is mostly descriptive and there is few insight by the author about what should be the future work, apart from "we need a better understanding".


Minor details:

Page 1: "can yield very strong performance on NLP tasks" is a very busy way to express the fact that Sentence Encoders work well in practice.

The field evolves quickly and ELMO has now a competitive models called BERT (arXiv.org > cs > arXiv:1810.04805). I understand that the results of the current papers would hove to be re-run on all these tasks, but I'm afraid the current paper will have a limited impact if it does not use the most effective method at the date of publication...

---

> ### Author Response · Authors · 2018-11-08
> **Author response**
>
> Thanks for your review! We agree that we have presented informative results on a few questions surrounding pretraining. We agree with some of your additional points, but we strongly disagree with your implication they make this paper not worth publishing.
>
> We don’t “believe that ELMO is the best contextual language model”, as you suggest. _Best_ is subjective, but we note on page 5 that Radford et al.’s Transformer model outperforms ELMo on GLUE.
>
> We focus on ELMo because it represented the state of the art as of spring, when we started building out the infrastructure for our experiments. Given the scale of this experiment, we would not have had time to conduct a similar analysis that follows the model and methods of Radford et al. (or BERT, which was released **after the ICLR deadline**), but we agree that a similar such analysis would be worthwhile.
>
> We also think that research on ELMo-style models would be worthwhile even if these fine-tuning based systems had come out earlier. ELMo’s pretrain-and-freeze approach to sentence representations is already being widely deployed (see Kitaev and Klein on parsing, or Gehrmann on summarization, for example), and it is likely the most practical approach if one wants to deploy a full NLP toolkit in a setting like a mobile phone where one cannot store a full separate fine-tuned 100MB-1GB encoder file for each task model that the toolkit provides.
>
> Further, many of our experiments on supervised pretraining do not involve ELMo, and we argue that the paper still offers more than enough worthwhile results even if you do not accept that ELMo is in any way worth studying.
>
> Zooming out a bit, though, we *strongly* disagree with the premise that it’s not worth publishing a paper if the baseline system does not represent the state of the art as of *publication* time, or that such papers cannot have a substantial impact. That would rule out most experimental papers, and it would rule out _any_ time-consuming analysis on exactly those topics where analysis would be most useful.

---

### Meta-Review · Area_Chair1 · 2018-12-14
**perhaps not strong novelty but interesting insights based on extensive experiments on ELMO**

**Recommendation:** Reject
**Confidence:** 3

**Metareview:**

This paper presents an extensive empirical study to sentence-level pre-training. The paper compares pre-trained language models to other potential alternative pre-training options, and concludes that while pre-trained language models are generally stronger than other alternatives, the robustness and generality of the currently available method is less than ideal, at least with respect to ELMO-based pretraining.

Pros:
The paper presents an extensive empirical study that offers new insights on pre-trained language models with respect to a variety of sentence-level tasks.

Cons:
The primarily contributions of this paper is empirical and technical novelty is relatively weak. Also, the insights are based just on ELMO, which may have a relatively weak empirical impact. The reviews were generally positive but marginally positive, which reflect that insights are interesting but not overwhelmingly interesting. None of these is a deal-breaker per say, but the paper does not provide sufficiently strong novelty, whether based on insights or otherwise, relative to other papers being considered for acceptance.

Verdict:
Leaning toward reject due to relatively weak novelty and empirical impact.

Additional note on the final decision:
The insights provided by the paper are valuable, thus the paper was originally recommended for an accept. However, during the calibration process across all areas, it became evident that we cannot accept all valuable papers, each presenting different types of hard work and novel contributions. Consequently, some papers with mostly positive (but marginally positive) reviews could not be included in the final cut, despite their unique values, hard work, and novel contributions.

---

> ### Author Response · Authors · 2018-12-21
> **Grumbly reply**
>
> I know that this is a moot point, but I do want to leave a brief response for the record.
>
> I sincerely hope that the stated reasons for rejection are not the real reasons. I won't claim that this paper deserved to get in, but I will claim that the stated reasons—that the paper presents new analysis and system-comparison results (rather than new methods) based on a several-months-old baseline—set very a worrying precedent.
>
> This would rule out _any_ time-consuming analysis on exactly those topics where analysis would be most useful.
>
> – SB

---

> > ### Public Comment · (anonymous) · 2019-03-08
> > **Subjective decision of the area chairs????**
> >
> > Additional note on the final decision is shocking!